# Corporate social responsibility and firm performance nexus: Moderating role of CEO chair duality

**Wasim Nasir[1], Arshad Hassan [1], Mushtaq Hussain Khan [2]***

**1** Faculty of Management & Social Sciences, Capital University of Science and Technology, Islamabad, Pakistan, **2** Cardiff School of Management, Cardiff Metropolitan University, Cardiff, United Kingdom

* MKhan3@cardiffmet.ac.uk

**Data Availability Statement:** The dataset used in this study has been uploaded as Supporting information file.

**Funding:** This research is funded by Cardiff Metropolitan University, UK under the PLOS

## Abstract

This study aims to explore the link between corporate social responsibility (CSR) and firm performance in the presence of the moderating role of CEO chair duality. It is widely believed that CSR initiatives and firm performance are largely influenced by psychological factors and the behavior of the decision maker (manager/CEO). Hence, CEO chair duality may play an instrumental role in shaping CSR initiatives to enhance firm performance. For empirical investigation, the study used the dynamic panel data method with generalized method of moment (GMM) parameters. The study considered 131 firms listed on the Pakistan Stock Exchange (PSX), yielding 1508 firm-year observations, over the period 2006 to 2020. Our results reveal that the impact of CSR on book-based and market-based measures differs due to the asymmetry of information in the market. The market discounts CEO chair duality due to the concentration of power and translates it into negative impact of CSR on firm performance. Thus, firms should not only improve CSR activities but also take steps to reduce asymmetry in markets because the impact on book-based measures and market-based measures of performance are not consistent. Society should also play a role to convince firms in a better way to take CSR initiatives. The perception of transparency should also be improved as CEO chair duality is being negatively seen by the market.

## Introduction

Corporate social responsibility (CSR) plays a key role in reshaping the corporate landscape. Corporate social responsibility is an instrument that assists firms in incorporating their voluntary social and environmental commitments into their operations and interactions with stakeholders. It is not just following regulations with a minimal approach. It is one step forward toward involving in answering the social needs of the stakeholders. This requires resource allocation in the development of human and environmental capital. Therefore, companies go beyond the minimum regulatory responsibilities and synchronize their economic interest with social and environmental interests. Therefore, companies think and exhibit socially responsible behavior for both moral and practical business motives. The supporters of socially responsible behavior highlight the benefits that companies can derive in the form of improved

Institutional Account Program. No additional external funding was received for this study. The funders had no role in study design, data collection and analysis, decision to publish, or preparation of the manuscript.

**Competing interests:** The authors have declared that no competing interests exist.

financial performance. This stream of literature opens the doors for the area of research that connects business and society. This area investigates the relationship between corporate social conduct and firm performance in the context of both the corporation's stakeholders and society.

Existing studies, for instance, McWilliams and Siegel [1] discuss and testify to the dichotomy between corporate social responsibility and firm performance but there is no consensus about the positive, negative, or no relationship. There are many reasons for such a mixed result. Some of these lies in the imperfections regarding the measurement of financial performance and corporate social responsibility, the omission of variables, confusion about the direction of causality, the lack of rigor in the statistical approach used, and inconsistency in the underpinning theory [2]. The debate has inconsistent arguments. The proponents of corporate social responsibility state that CSR has a positive impact on financial performance. The critics argue CSR involves unnecessary costs that reduce profitability. Therefore, the literature in this domain is very diversified. There is a conflicting theoretical framework. The debate has two major perspectives. The studies that consider CSR assignments as an investment and studies that consider these as an agency cost.

Freeman [3] was the first to introduce the concept of stakeholder theory. This theory is a fundamental perspective used to conceptualize the connection between corporate social responsibility (CSR) and performance. Research studies that adopt the stakeholder theory viewpoint investigate the correlation between stakeholder management and its influence on the performance of a company [4]. Jones [5] developed an instrumental theory that combines stakeholder theory, economic theory, and ethical standards.

The study suggests that markets are competitive and discipline the behavior of firms through a pricing mechanism, so firms are forced to exercise instrumental stakeholder management to attain a competitive edge. The positive relationship between CSR and firm performance has been discussed under the social impact hypothesis which asserts that supporters of the stakeholder theory believe that favorable social performance in the form of meeting the expectations of stakeholders leads to favorable firm performance and vice versa [6].

According to Friedman [7], companies with robust social credentials tend to see a decrease in their stock prices compared to the market average. The trade-off hypothesis, introduced by Aupperle et al. [8] suggests that socially responsible activities such as corporate philanthropy, environmental initiatives, and community development may require significant resources from the firm, potentially putting it at a disadvantage compared to less socially active firms. As a result, a company's increased level of social performance may result in lower financial performance when compared to its competitors. According to the "managerial opportunism" hypothesis, there is a close link between the pursuit of private managerial objectives within compensation schemes and short-term profit as well as the behavior of stock prices. This connection could result in an adverse correlation between financial and social outcomes. When a company is performing well financially, managers may prioritize their own short-term gains by cutting back on social expenses. Conversely, when financial performance is poor, managers may attempt to offset and rationalize their unsatisfactory results by engaging in conspicuous social initiatives.

The first set of studies reports a positive correlation between corporate social responsibility (CSR) and financial performance. Wu [9], Peiris and Evans [10], Gimeno-Arias et al. [11], Palacios-Manzano et al. [12], and Ortiz-Martínez et al. [13] belong to this group, as they find a significant positive association between CSR and financial performance. Margolis et al. [14] suggest that CSR has a positive impact on profits, as measured by book-based and market-based proxies. Other studies have shown that companies with higher CSR scores tend to perform better than those with lower scores [15, 16]. The second set of studies argues that CSR

has a negative influence on firm performance. Fisher-Vanden and Thorburn [17] propose that the announcement of companies joining an environmentally friendly program may trigger a negative market reaction due to the expectation of adverse effects on performance. However, the empirical evidence on whether CSR practices have a positive, negative, or no impact on financial performance remains inconclusive, reflecting the divided opinions on the empirical results. Wang [18] provides a systematic review of the link between CSR and corporate financial performance, finding a positive and significant relationship between the two. The meta-analysis supports the notion that CSR enhances firms' financial outcomes. This study offers insights into the direction of the relationship between CSR and financial performance, concluding that financial performance is related to past social responsibility, but the reverse direction is not supported. Wang et al. [18] also suggest that CSR has different implications for financial investors and other stakeholders, and its effect on financial outcomes varies. Recently, León-Gómez et al. [19] and Santos-Jaén et al. [20] investigate the role of CSR in the relationship between information and communication technologies adoption (ICT) and SMEs performance in the hotel industry. Their findings suggest that considering ICT as source of competitive edge encourages the implementation of CSR practices, which, in turn, enhances firm performance in hotel industry. Likewise, Becerra-Vicario et al. [21] test the mediating role of CSR in the association between innovation and SMEs performance in the industrial sector and provide the evidence of the impact of CSR on the link between innovation and SMEs performance.

Corporate governance is a mechanism that uses the forum of the board of directors to address agency-related problems. In their study, Achim et al. [22] present findings that suggest corporate governance has a beneficial effect on firm performance in the Romanian market. Specifically, factors such as the size and independence of the board, as well as whether the CEO and Chairman positions are held by the same person, may influence both corporate social responsibility (CSR) and the financial performance of firms. Cordeiro et al. [23] find a U-Type non-linear relationship between CSR engagement and financial performance in family-owned firms in India. The larger board has more time and expertise to oversee the affairs of the companies, therefore the decision of large boards can reinforce the impact of CSR activities on a firm financial performance more than those of smaller boards. The independent board has more focused on CSR activities because of less financial interest in the companies. Recently, Javeed and Lefen [24] explore the moderating role of CEO power and ownership on the CSR-firm performance nexus. They used CEO compensation as a measure of CEO power; however, our study focuses on the moderating role of CEO duality. We suggest that boards with a high proportion of independent directors strengthen the impact of CSR on financial performance. The CEO/Chairman duality concentrates the power in the hands of one person and this strong power enhances the capability of the CEO to influence decisions. Therefore, CEO duality deems to be a moderator which may have a significant effect on the relationship between CSR activities and a firm financial performance.

## Data and methodology

The sample of the study consists of 131 companies listed on Pakistan Stock Exchange. The companies are selected from the non-financial sector based on market capitalization. The sample period is 2006 to 2020. The reason for taking a sample from 2006 is the adoption of a code of corporate governance in 2005 in Pakistan.

Firm performance is measured through Return on Asset (ROA) and Tobin Q. Corporate social performance is measured through the percentage of profit allocated for social activities. The firm-specific variables include sales growth, firm size, a book-to-market ratio (BMR), and

**Table 1. Measurement of variables.**

| Variables | Abbreviation | Formula |
|---|---|---|
| Return on Assets | ROA | Net Profits/Total Assets |
| Market to Book Ratio | BMR | Market Value Per Share/Book Value Per Share |
| Tobin Q | TQ | Market Value of Firm/Total Asset Value of Firm |
| Corporate Social Responsibility | CSR | Funds Allocated for CSR/Total Profit |
| Sales Growth | SG | $S_t - S_{t-1}/ S_{t-1}$ |
| Leverage | LEVE | Debt/Equity |
| Market Capitalization | MCAP | Market Price × Number of Outstanding Shares |
| Size of Board | BS | ln (No. of Directors) |
| Board Independence | BIP | No. of Non-Executive Directors/Total Number of Directors |
| CEO Duality | CEOD | "1", if the CEO and Chairman is the same person "0", otherwise |

leverage. The board attributes used include the size of the board, board independence, and CEO/Chair duality.

To enhance the reliability of the findings and tackle the problem of endogeneity in the panel dataset, the study utilizes the GMM method which involves the use of instrumental variables. By doing so, the methodology aims to minimize the link between the stochastic error terms and the explanatory variables, thus enhancing the robustness of the results. The Econometric Model is presented below.

$$FP_{it}(t, h) = \beta_o + \beta_1 CSR_{it} + \beta_2 BS_{it} + \beta_3 BIP_{it} + \beta_4 CEOD_{it} + \beta_5 COED_{it} \times CSR_{it} + \sum_{i=0}^{n} \delta_i x_{it}$$
$$+ \mu_{it} \tag{1}$$

The study considers Arellano and Bond [25] method that indicates that the estimation procedure uses the first difference data. It also uses Arellano and Bover [26] method to perform Orthogonal deviations to remove the individual effects. The variables are measured as detailed below in Table 1.

## Results and discussion

The statistical behavior of data is examined through descriptive statistics. Table 2 exhibits the measures of central tendency, measures of dispersion, and location parameters. The average

**Table 2. Descriptive statistics.**

| Variables | Mean | Median | Max. | Min. | Std. Dev. | Skewness | Kurtosis | Prob. |
|---|---|---|---|---|---|---|---|---|
| CSR | 0.062 | 0.012 | 10.09 | 0.000 | 0.289 | 28.2 | 963.3 | 0.00 |
| ROA | 0.068 | 0.059 | 0.463 | -1.174 | 0.089 | -1.124 | 27.71 | 0.00 |
| TQ | 1.476 | 1.047 | 25.43 | 0.156 | 1.571 | 5.959 | 58.60 | 0.00 |
| SG | 0.128 | 0.108 | 3.556 | -0.909 | 0.298 | 3.096 | 27.70 | 0.00 |
| LEVE | 0.537 | 0.547 | 7.750 | 0.032 | 0.280 | 11.74 | 296.5 | 0.00 |
| LOG(MCAP) | 17.30 | 17.28 | 23.14 | 11.44 | 2.100 | 0.032 | 2.458 | 0.00 |
| BMR | 0.161 | 0.091 | 4.105 | -0.563 | 0.253 | 6.252 | 67.72 | 0.00 |
| BS | 8.306 | 8.000 | 15.00 | 5.000 | 1.760 | 1.585 | 6.046 | 0.00 |
| BIP | 0.258 | 0.143 | 1.000 | 0.045 | 0.201 | 1.589 | 4.685 | 0.00 |
| CEOD | 0.214 | 0.000 | 1.000 | 0.000 | 0.410 | 1.397 | 2.951 | 0.00 |

**Table 3. Impact of CSR and ROA with moderating role of CEO duality.**

| | Static Model | | Dynamic Model | |
|---|---|---|---|---|
| | **OLS** | **Fixed Effect** | **Orthogonal Deviations** | **First Difference** |
| **Variables** | **ROA** | **ROA** | **ROA** | **ROA** |
| ROA $_{t-1}$ | | | 0.0636** (0.0181) | 0.0595** (0.0154) |
| ROA $_{t-2}$ | | | 0.1004** (0.0116) | 0.0942** (0.0111) |
| TCSR | 0.0179** (0.0063) | -0.0074 (0.0040) | 0.0110** (0.0030) | 0.0129** (0.0031) |
| SG | 0.0302** (0.0038) | 0.0265** (0.0042) | -0.0034 (0.0031) | -0.0043 (0.0032) |
| LEVE | -0.1493** (0.0047) | -0.1521** (0.0148) | -0.5803** (0.0271) | -0.6081** (0.0279) |
| LOG (MCAP) | 0.0062** (0.0006) | 0.0089** (0.0015) | -0.0059* (0.0028) | -0.0063* (0.0027) |
| BMR | -0.0199** (0.0038) | -0.0076 (0.0049) | 0.0202 (0.0264) | 0.0199 (0.0250) |
| BIP | 0.0027 (0.0042) | 0.0166** (0.0069) | 0.0437 (0.0337) | 0.0304 (0.0369) |
| LOG (BS) | 0.0315** (0.0059) | 0.0140 (0.0157) | 0.3036** (0.1152) | 0.3095** (0.1092) |
| CEOD | -0.0079** (0.0022) | 0.0081 (0.0045) | -0.0329 (0.0223) | -0.0314 (0.0221) |
| CEOD*TCSR | 0.0019 (0.0122) | -0.0163** (0.0057) | -0.3172** (0.1243) | -0.3354** (0.1242) |
| Constant | -0.0347** (0.0127) | -0.0432 (0.0442) | | |
| Adjusted R$^2$ | 0.5172 | 0.7848 | | |
| F-Statistic | 180.2648 | 41.3727 | | |
| Prob(F-Statistic) | 0.0000 | 0.0000 | | |
| D-W Stat | 1.0403 | 1.6171 | | |
| J-Statistic | | | 52.9049 | 51.3714 |
| Prob(J-Statistic) | | | 0.1432 | 0.1786 |
| Instrument Rank | | | 54 | 54 |

**Note:** The figures in parentheses are standard errors.

* and ** are the level of significance at 95% level and 99%

return on assets is 6.8% per annum with an average variation of 8.9% per annum. Pakistani companies on average spent 0.06% of the profit on social activities which is a small amount. The sales growth rate is 12.8% whereas the average variation in growth rate is 29.8%. The average leverage is 53.7%. The average book-to-market ratio is 0.16 which is low indicating higher market prices.

A birds-eye view of board characteristics provides that the average board size is 8 and the largest board comprises 15 members and the smallest board size comprises 5 members. On Average 25% of the boards are independent. However, most independent board comprises 100%, independent members. These are generally boards of state-owned companies where government nominates the members. Twenty-one percent of companies have a chairman who is the CEO of the company too. The data is positively skewed, and kurtosis is more than 3 for most of the variables that show the non-normality of data.

Table 3 exhibits the estimates of coefficients for the econometric model explaining the connection between CSR and return on assets with the moderating role of CEO Chair duality. The results of the Static and Dynamic models under different assumptions are reported below.

The fixed-effect model based on unbalanced panel data shows that CSR has an insignificant impact on return on the asset at a 95% level of significance. The Generalized Method of Moment Model shows that CSR has a statistically significant and positive effect on return on the asset at the 99% significance level. The static Fixed effect model is weaker due to the presence of lagged relationship and simultaneity so dynamic panel data analysis is performed by using the Generalized method of moments. The findings of the study proposed by GMM

provide that there exists a significant positive relationship between CSR and ROA under both assumptions.

The findings suggest that higher debt levels negatively affect profitability, as evidenced by the significant negative coefficient. Additionally, firm size is a crucial factor in explaining firm performance due to its correlation with economies of scale, as noted by Dang et al. [27]. The results related to firm size, however, are varied and not surprising, given that smaller firms often have more growth potential than larger ones. This is due to the curvilinear relationship between firm size and performance, as noted by Lin et al. [28].

The presence of independent board members has a notable and favorable impact on a company's performance. This is because it enhances transparency in the decision-making process, which leads to increased financial benefits. On the other hand, having a CEO with dual roles has a detrimental but unimportant effect on company performance. Regarding the analysis of moderation, the interaction term is the most significant variable. The negative coefficient of CSR*CEOD, which is statistically significant at the 1% level, suggests that, for companies where the CEO holds dual power, the average improvement in performance controlled by CSR is lower than that of other firms, even when accounting for other factors. These findings validate the notion that CEO power has negative consequences for the relationship between CSR and company performance, especially for organizations where the CEO has greater power.

The findings of the two models indicate that there is a positive relationship between allocations for social responsibility and firm performance. As such, owners and stakeholders stand to benefit from social activities, as the associated financial gains tend to improve with CSR initiatives. Investing in social activities can also lead to positive market feedback, significant net profit increases, and greater financial growth stability, as noted by [29]. Therefore, the association between CSR and firm performance is found to be positive. The concentration of power in the hands of one person is discounted and the impact of CSR on return on asset dilutes.

Table 4 exhibits the estimates of coefficients for the model explaining the connection between CSR and Tobin Q with the moderating role of CEO Chair duality. The results of the Static and Dynamic models are reported under different assumptions are reported below.

The fixed-effect model shows that CSR has a significant and negative impact on Tobin Q at a 95% level of significance. However, the issue of autocorrelation, simultaneity, and endogeneity is observed. The Generalized Method of Moment Model shows that CSR has a statistically significant and negative effect on Q at the 99% significance level. The findings of the study by GMM provide that there exists a significant negative relationship between CSR and Q under both assumptions. The performance of big companies seems better than smaller companies.

The book-to-market ratio which is a measure of the market expectation of growth is also significant and positive. The firms with high BMR are better than firms with lower BMR. Companies with big board sizes have a negative influence on performance. The large board may have a large set of expertise but still, have a problem in getting a consensus that liquidates the decisions. CEO duality has a negative and statistically significant impact on performance, and it further strengthens when used as a moderator between CSR and Tobin Q. This shows that in the case of CEO duality, CSR's impact on firm performance is considered negative which is in line with agency theory. The market may consider that decisions are taken based on personal preference and not based on the institutional perspective. The proponents of agency theory further argue that the CEO and Chairman of the board should be separated because a single person can dominate company affairs and decision making which can lead to the concentration of power in one person and promote managerial opportunism which will affect performance.

**Table 4. Impact of CSR and TQ with moderating role of CEO duality.**

| | Static Model | | Dynamic Model | |
|---|---|---|---|---|
| | OLS | Fixed Effect | Orthogonal Deviations | First Difference |
| Variables | TQ | TQ | TQ | TQ |
| TQ $_{t-1}$ | | | 0.2243** (0.0124) | 0.2284** (0.0122) |
| TQ $_{t-2}$ | | | -0.2277** (0.0128) | -0.2256** (0.0123) |
| TCSR | 0.1149** (0.0371) | -0.1803** (0.0442) | -1.3256** (0.2977) | -1.3325** (0.3281) |
| SG | 0.0208 (0.0222) | -0.0125 (0.0296) | 0.1804 (0.1689) | 0.2067 (0.1653) |
| LEVE | 0.3359** (0.0385) | 0.5351** (0.0978) | -0.2686 (0.1711) | -0.2725 (0.1799) |
| LOG (MCAP) | 0.0229** (0.0048) | 0.2401** (0.0478) | 0.5942** (0.0801) | 0.5629** (0.0839) |
| BMR | -0.4112** (0.0742) | 0.1247 (0.0967) | 3.4706** (0.6775) | 3.3778** (0.6889) |
| BIP | | 0.0708 (0.0732) | -0.7084 (1.0829) | -0.8872 (1.2406) |
| LOG (BS) | 0.0986** (0.0219) | -0.7367** (0.1803) | -14.8787** (1.3259) | -14.2814** (1.2253) |
| CEOD | -0.0258** (0.0109) | -0.0117 (0.0301) | -1.2553** (0.3706) | -1.3901** (0.4428) |
| CEOD*TCSR | -0.2488** (0.0806) | -0.3011 (0.2359) | -12.2750* (6.1896) | -11.6788* (6.1974) |
| C | 0.2148** (0.0905) | -1.4418 (0.8628) | | |
| Adjusted R$^2$ | 0.3489 | 0.6899 | | |
| F-Statistic | 101.9130 | 25.6379 | | |
| Prob (F-Statistic) | 0.0000 | 0.0000 | | |
| D-W Stat | 0.9394 | 1.4631 | | |
| J-Statistic | | | 48.5307 | 46.8650 |
| Prob (J-Statistic) | | | 0.2598 | 0.3169 |
| Instrument Rank | | | 54 | 54 |

**Note:** The figures in parentheses are standard errors.

* and ** are the level of significance at 95% level and 99%

The findings of the study suggest that CSR has a significant and positive relationship with firm performance captured through book-based whereas it has a significant and negative relationship with firm performance captured through market-based measures. This pattern indicates that CSR perception can be differently priced by the market. The concentration of power in the hand of one person is negatively interpreted by the market and it even reduces the impact of good work.

The debate regarding the financial advantages of corporate social responsibility (CSR) is heavily influenced by the context in which it is considered. Wealth maximization theory findings indicate that corporate social performance (CSP) contradicts a firm's goal of maximizing its value as it draws resources away from its core operations. This perspective is supported by several scholars, including Friedman [7], Aupperle et al. [8], McWilliams and Siegel [30], and Jensen [31]. Financial stakeholders may perceive CSP as a hindrance to a firm's financial success and consequently disregard its importance.

Conversely, stakeholder theory proposes that CSP contributes to a positive external perception of the organization, resulting in supportive behavior and potentially leading to increased sales and higher bottom lines [32]. To resolve this inconsistency, Hillman and Keim [33] suggest that the association between CSR and financial performance is dependent on the perceptions of both financial and common stakeholders.

## Conclusions and policy recommendations

This study provides empirical evidence about the relationship between CSR and firm performance. The findings of the study reveal that CSR and return on assets have a positive and

statistically significant relationship implying that CSR can enhance financial gains and confidence of the stakeholders which is consistent with the stakeholder theory. However, it is interesting that the relationship between CSR and firm performance is negative when the market-based measure of performance is used. This shows that market response may differ from information disclosed in financial reports. These findings are consistent with the argument of Lange et al. [34], "an organization's external observers have varying interests and therefore are attuned to different valued organizational outcomes" (p. 164). For example, environmental activists and financial investors may view differently when valuing a firm's commitment to adapting the social norms.

Furthermore, the study finds that the interaction between CSR and CEO duality is negatively related to return on asset and Tobin Q. It means that the concentration of power in one person negatively influences the firm. It seems that it is considered an indicator of weak transparency and accountability mechanism. The other reason may be that most of the firms are family-owned in Pakistan and powerful CEOs may use their power for personal objectives.

Based on the important insights mentioned above, this study offers the following recommendations to improve the CSR initiatives in Pakistan. Importantly, the focus of firms should not be only to improve CSR activities but also to take steps to reduce asymmetry in markets because the impact on book-based performance and market-based performance is inconsistent. Society should also play a role to convince firms in a better way to take CSR initiatives in Pakistan. The perception of transparency should also be improved as CEO chair duality is being negatively seen by the market. Firms should not only focus on their business operations but also consider the concerns and needs of the communities in which they operate and strive to develop effective solutions for addressing community issues and creating awareness in society about its role in a social cause. This study encourages investors, owners, and shareholders, to contribute more to CSR initiatives. For future research, the ownership, Board composition, financing structure, and compliance requirements may be the main concerns for considering as a moderator. This study is also applicable to firms in countries where an entrepreneurial corporate form of business is present.

## Supporting information

**S1 Dataset.**
(XLSX)

## Author Contributions

**Conceptualization:** Wasim Nasir, Arshad Hassan.

**Formal analysis:** Arshad Hassan.

**Investigation:** Wasim Nasir.

**Methodology:** Arshad Hassan.

**Supervision:** Arshad Hassan.

**Validation:** Wasim Nasir, Mushtaq Hussain Khan.

**Writing – original draft:** Wasim Nasir.

**Writing – review & editing:** Mushtaq Hussain Khan.

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
