## [Decision Letter · Decision Letter 0]

24 May 2023

PONE-D-23-13630Corporate Social Responsibility and Firm Performance Nexus: Moderating Role of CEO Chair DualityPLOS ONE

Dear Dr. Khan,

Thank you for submitting your manuscript to PLOS ONE. After careful consideration, we feel that it has merit but does not fully meet PLOS ONE’s publication criteria as it currently stands. Therefore, we invite you to submit a revised version of the manuscript that addresses the points raised during the review process.

We look forward to receiving your revised manuscript.

Kind regards,

José Manuel Santos Jaén

Academic Editor

PLOS ONE

Journal Requirements:

Additional Editor Comments: 

In order to publish the article, the authors must take into account the indications made by the reviewer.

Reviewers' comments:

Reviewer's Responses to Questions

**Comments to the Author**

1. Is the manuscript technically sound, and do the data support the conclusions?

Reviewer #1: Yes

2. Has the statistical analysis been performed appropriately and rigorously? 

Reviewer #1: Yes

3. Have the authors made all data underlying the findings in their manuscript fully available?

Reviewer #1: Yes

4. Is the manuscript presented in an intelligible fashion and written in standard English?

Reviewer #1: Yes

5. Review Comments to the Author

Reviewer #1: This article is of great interest to the empirical literature published so far, as it deals with a highly topical and relevant subject. The authors have made a great effort, and I congratulate them on their work. The article is solid and responds to the stated objectives. The conclusions contribute to future articles that may be published in this area. However, the literature review section should be updated with references closer to the present day. That is why I suggest these articles to the authors, which I believe could be of great help to you:

Gimeno-Arias, F., Santos-Jaén, J. M., Palacios-Manzano, M., & Garza-Sánchez, H. H. (2021). Using PLS-SEM to analyze the effect of CSR on corporate performance: The mediating role of human resources management and customer satisfaction. An empirical study in the Spanish food and beverage manufacturing sector. Mathematics, 9(22), 2973

Palacios-Manzano, M., León-Gomez, A., & Santos-Jaén, J. M. (2021). Corporate Social Responsibility as a Vehicle for Ensuring the Survival of Construction SMEs. The Mediating Role of Job Satisfaction and Innovation. IEEE Transactions on Engineering Management.

León-Gómez, A., Santos-Jaén, J. M., Ruiz-Palomo, D., & Palacios Manzano, M. (2022). Disentangling the impact of ICT adoption on SMEs performance: the mediating roles of corporate social responsibility and innovation. Oeconomia Copernicana, 13(3), 831–866. doi:10.24136/oc.2022.024

Santos-Jaén, J. M., León-Gómez, A., Ruiz-Palomo, D., García-Lopera, F., & Valls Martínez, M. del C. (2022). Exploring Information and Communication Technologies as Driving Forces in Hotel SMEs Performance: Influence of Corporate Social Responsibility. Mathematics, 10(19), 3629. https://doi.org/10.3390/math10193629

Ortiz-Martínez, E., Marín-Hernández, S., & Santos-Jaén, J.-M. (2023). Sustainability, corporate social responsibility, non-financial reporting and company performance: Relationships and mediating effects in Spanish small and medium sized enterprises. Sustainable Production and Consumption, 35, 349–364. https://doi.org/10.1016/j.spc.2022.11.015

Becerra-Vicario, R., Ruiz-Palomo, D., León-Gómez, A., & Santos-Jaén, J. M. (2023). The Relationship between Innovation and the Performance of Small and Medium-Sized Businesses in the Industrial Sector: The Mediating Role of CSR. Economies, 11(3), 92. https://doi.org/10.3390/economies11030092

6. PLOS authors have the option to publish the peer review history of their article (what does this mean?). If published, this will include your full peer review and any attached files.

Reviewer #1: No

---

## [Author Response · Author response to Decision Letter 0]

6 Jul 2023

RESPONSE TO THE VALUABLE COMMENTS OF THE ACADEMIC EDITOR AND REVIEWERS

Corporate social responsibility and firm performance nexus: moderating role of CEO chair duality

Reference: PONE-D-23-13630

The authors would like to thank the Editor, for the opportunity to revise the original manuscript. We also thank the anonymous reviewers for providing us with valuable comments and suggestions, which have improved the quality of the paper. We have examined all the general and specific comments provided by the reviewer and accordingly made the necessary changes to meet their suggestions. We have presented below our responses to reviewer’s query.

We hope that these revisions have improved the quality of the present version of the manuscript and will earn the satisfaction of the reviewer.

RESPONSE TO ACADEMIC EDITOR

Comment # 1: 

Response:

Thank you. Three separate files have been uploaded as per your suggestion. In the case of 'Revised Manuscript with Track Changes', changes have been highlighted using coloured text.

Comment # 2: 

Response:

In the revised version, the PLOS ONE's style guide has been followed.

Comment # :3 

Please ensure that you include a title page within your main document. Could you therefore please include the title page into the beginning of your manuscript file itself, listing all authors and affiliations.

Response:

Title page has been included into the beginning the manuscript file as per instructions.

Comment # 4: 

We note that you have indicated that data from this study are available upon request. PLOS only allows data to be available upon request if there are legal or ethical restrictions on sharing data publicly. In your revised cover letter, please address the following prompts:

b) If there are no restrictions, please upload the minimal anonymized data set necessary to replicate your study findings as either Supporting Information files or to a stable, public repository and provide us with the relevant URLs, DOIs, or accession numbers. 

Response:

The dataset used in this study has been uploaded as Supporting Information file.

Comment # 5: 

PLOS requires an ORCID iD for the corresponding author in Editorial Manager on papers submitted after December 6th, 2016. Please ensure that you have an ORCID iD and that it is validated in Editorial Manager. To do this, go to ‘Update my Information’ (in the upper left-hand corner of the main menu), and click on the Fetch/Validate link next to the ORCID field. 

Response:

Thank you for providing guidance. The ORCID iD of corresponding author has been linked with the Editorial Manager.

Comment # 6: 

Please review your reference list to ensure that it is complete and correct. 

Response:

In the revised version, reference list and in-text citations have been carefully matched and corrected. Moreover, to manage the consistency in reference list and in-text citations, EndNote has been used.

RESPONSE TO REVIEWER # 1

Comment # 1: 

This article is of great interest to the empirical literature published so far, as it deals with a highly topical and relevant subject. The authors have made a great effort, and I congratulate them on their work. The article is solid and responds to the stated objectives. The conclusions contribute to future articles that may be published in this area. 

Response:

We would like to thank you for this positive comment. 

Comment # 2: 

The literature review section should be updated with references closer to the present day. That is why I suggest these articles to the authors, which I believe could be of great help to you:

1. Gimeno-Arias, F., Santos-Jaén, J. M., Palacios-Manzano, M., & Garza-Sánchez, H. H. (2021). Using PLS-SEM to analyze the effect of CSR on corporate performance: The mediating role of human resources management and customer satisfaction. An empirical study in the Spanish food and beverage manufacturing sector. Mathematics, 9(22), 2973

2. Palacios-Manzano, M., León-Gomez, A., & Santos-Jaén, J. M. (2021). Corporate Social Responsibility as a Vehicle for Ensuring the Survival of Construction SMEs. The Mediating Role of Job Satisfaction and Innovation. IEEE Transactions on Engineering Management.

3. León-Gómez, A., Santos-Jaén, J. M., Ruiz-Palomo, D., & Palacios Manzano, M. (2022). Disentangling the impact of ICT adoption on SMEs performance: the mediating roles of corporate social responsibility and innovation. Oeconomia Copernicana, 13(3), 831–866. doi:10.24136/oc.2022.024

4. Santos-Jaén, J. M., León-Gómez, A., Ruiz-Palomo, D., García-Lopera, F., & Valls Martínez, M. del C. (2022). Exploring Information and Communication Technologies as Driving Forces in Hotel SMEs Performance: Influence of Corporate Social Responsibility. Mathematics, 10(19), 3629. https://doi.org/10.3390/math10193629

5. Ortiz-Martínez, E., Marín-Hernández, S., & Santos-Jaén, J.-M. (2023). Sustainability, corporate social responsibility, non-financial reporting and company performance: Relationships and mediating effects in Spanish small and medium sized enterprises. Sustainable Production and Consumption, 35, 349–364. https://doi.org/10.1016/j.spc.2022.11.015

6. Becerra-Vicario, R., Ruiz-Palomo, D., León-Gómez, A., & Santos-Jaén, J. M. (2023). The Relationship between Innovation and the Performance of Small and Medium-Sized Businesses in the Industrial Sector: The Mediating Role of CSR. Economies, 11(3), 92. https://doi.org/10.3390/economies11030092

Response:

Thank you for constructive comment and for suggesting important studies. In the revised version, suggested studies have been included.

Once again, thank you for the detailed and insightful comments. We are, of course, open to any additional suggestions.

---

## [Editor Report · Decision Letter 1]

10 Jul 2023

Corporate social responsibility and firm performance nexus: moderating role of CEO chair duality

PONE-D-23-13630R1

Dear Dr. Khan,

We’re pleased to inform you that your manuscript has been judged scientifically suitable for publication and will be formally accepted for publication once it meets all outstanding technical requirements.

Kind regards,

José Manuel Santos Jaén

Academic Editor

PLOS ONE

Additional Editor Comments (optional):

Having verified that the authors have implemented all the recommendations made by the reviewers, I proceed to accept the article.

Good work!
---

## [Editor Report · Acceptance letter]

25 Jul 2023

PONE-D-23-13630R1 

Corporate social responsibility and firm performance nexus: moderating role of CEO chair duality 

Dear Dr. Khan:

I'm pleased to inform you that your manuscript has been deemed suitable for publication in PLOS ONE. Congratulations! Your manuscript is now with our production department. 

Kind regards, 

on behalf of

Dr José Manuel Santos Jaén 

Academic Editor

PLOS ONE